# DON'T PRE-TRAIN, TEACH YOUR SMALL MODEL

## ABSTRACT

In this paper, we reconsider the question: *What is the most effective way to train a small model?* A standard approach is to train it from scratch in a supervised manner on the desired task for satisfactory results at a low cost. Alternatively, one can first pre-train it on a large foundation dataset and then finetune it on the downstream task to obtain strong performance, albeit at a much higher total training cost. Is there a middle way that balances high performance with low resources? We find the answer to be **yes**. If, while training from scratch, we regularize a small model to match an existing pre-trained one on the relevant subset of the data manifold, it can achieve *similar or better* performance than if it was completely pre-trained and finetuned. We accomplish this via a novel paradigm, inspired by knowledge distillation, that can take *any* public pre-trained model, *any* contrastive learning algorithm, and *any* public pre-trained text-to-image generative model to give small models an unprecedented boost in performance. The cost is only slightly higher than end-to-end supervised training on a desired task. In interest of the broader community, our work shows that advancing foundation models, contrastive learning, and pre-trained generative models *indirectly benefits* their less popular predecessor: small, task-specific models. We demonstrate the efficacy of our approach across 6 image recognition datasets, utilizing pre-trained convolution and attention-based teachers from public model hubs, an inexpensive yet flexible distillation algorithm derived from modern contrastive learning theory, and an out-of-the-box diffusion model that never touches the task dataset. Seeing as our method can hold its weight against, and often surpass, the pre-training regime, we refer to our paradigm as: **Don't Pre-train, Teach (DPT)**.

## 1 INTRODUCTION

Small machine learning models are incredibly useful. For those looking to save costs on memory and compute, reduce their energy footprint, or bring machine learning to edge devices such as mobile phones, small models that perform well on desired tasks are the way to go.

Recently, most attention has focused on developing larger models trained on larger datasets, dubbed "foundation models" (Radford et al., 2021; 2019; Brown et al., 2020; Chen et al., 2020a; Bommasani et al., 2021; Alayrac et al., 2022; Yuan et al., 2021). It is generally accepted, though the exact reason *why* still eludes us, that as the scales of both model and dataset increase, accuracy increases. At the same time, how have small models kept up in performance?

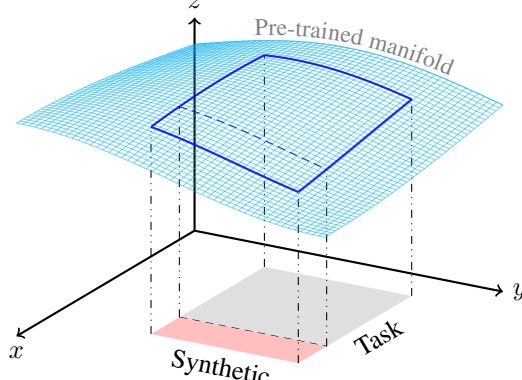

Figure 1: By only mimicking the relevant slice of a pre-trained manifold, a small model can achieve the *same or better* performance than if it had been fully pre-trained and finetuned. Adding synthetic samples leads to better generalization.

Various lines of research have tackled how to squeeze out maximal performance from small models, all targeting different points in the training process. Before training, neural architecture search (NAS) (Zoph & Le, 2017; Liu et al., 2019) can algorithmically find a small model design that should perform well on various tasks. After training,

one can prune or quantize (Jacob et al., 2017; Han et al., 2016) the model to reduce its size and keep, theoretically, the same performance. We focus on what happens in the middle: the training process. During training, knowledge distillation (KD) Bucilua et al. (2006); Hinton et al. (2015) can assist the model by having it also learn from a larger teacher model that is an expert at the task.

Our motivating problem comes from the classic strategy of *pre-training*: train the model on a large, diverse dataset such that it will become a generic feature extractor. Then, finetune it on downstream tasks, which usually leads to superior performance than if it had been trained on the task from scratch. The effectiveness of this approach is so widely observed that online hubs with pre-trained models are indispensable in modern machine learning applications. However, if one needs to use an architecture that is *not* available in these hubs, then they face a tradeoff: (a) either absorb the cost of pre-training and achieve great results or (b) train from scratch on just the desired task, sacrificing performance.

To help tackle this dilemma, we re-evaluate if pre-training is necessary for small models with the following insight: if the end goal is to maximize performance on a specific task, does a small model need a *comprehensive* feature backbone? Alternately, what if we teach it to behave like it was pre-trained and finetuned, but only on the relevant slice of knowledge (see Figure 1)?

In this paper, we show that, by leveraging the progress in foundation models, contrastive learning, and pre-trained generative models in a carefully constructed paradigm, small models can achieve and *surpass* pretrain-then-finetune performance without ever needing to touch a pre-training dataset. Our approach is simple yet effective: (1) using *any* publicly available teacher pre-trained on the appropriate foundation dataset, first finetune it for the desired task, then (2) train the small model in a supervised manner and regularize it to match the teacher via our novel (or possibly other) knowledge distillation loss on (3) the task dataset augmented with synthetic samples generated from publicly available pre-trained generative models. Our method can easily be applied to improve performance for any small model on any task, can be used in continual learning regimes (Li & Hoiem, 2017) if multiple tasks are desired, and fits in nicely between methods like NAS and pruning to fully utilize all the tools to maximize small model accuracy.

We test our approach on 6 visual recognition tasks, spanning both the data-abundant and data-limited regimes, the latter of which benefits the most from pre-training. The formulation of our KD loss as a contrastive objective allows our setup to be agnostic to the underlying teacher-student architectures. We demonstrate this by utilizing teachers representative of the main vision architectures, a Vision Transformer (Dosovitskiy et al., 2020) and 50-layer ResNet (He et al., 2016), to assist 2 small students, a MobileNetV2 (Sandler et al., 2018) and 18-layer ResNet. We also provide a cost analysis of our method: skipping pre-training leads to large resource benefits by saving weeks of compute. However, a nontrivial generation cost of hundreds of model evaluations per input is added given the state of the diffusion model we employ. Given the efficacy of our method, we refer to our proposed paradigm as **Don't Pre-train, Teach (DPT)**. Our code will be made publicly available upon acceptance.

Our contributions can be summarized as follows:

- We propose a competitive yet cheaper strategy to pre-training and finetuning for small models. By mimicking a pre-trained model on the relevant subset of the data manifold, one can get the accuracy benefits of pre-training without needing to actually do it. From a cost perspective, this enables us to sidestep the expensive pre-training process and still maintain high accuracies while using up to 94% less training time.

- We re-formulate knowledge distillation in terms of modern contrastive learning, allowing all algorithms from their literature to be freely utilized here. We illustrate this via a novel knowledge distillation algorithm based on the Alignment/Uniformity perspective of Wang & Isola (2020). Notably, our formulation enables it to be used for *any* architecture pairing and achieves competitive results with other state of the art methods, while being the cheapest contrastive-based distillation algorithm to date.

- Motivated by what we define in Section 3.3.1 as the *data gap*, we show that pre-trained diffusion models (Rombach et al., 2022) are beneficial in augmenting transfer datasets when doing KD, leading to significant increases in accuracy, especially in low-data regimes.

## 2 RELATED WORK

### 2.1 KNOWLEDGE DISTILLATION

Knowledge Distillation (KD) first appeared in the realm of ensembling methods (Dietterich, 2000), which takes multiple models trained on a task, combines their outputs, and averages them to provide a useful signal for a new model to learn from. Bucilua et al. (2006) were the first to formally apply this with $n \geq 1$ expert models but targeted a smaller model, thus considering their technique a form of model compression. Hinton et al. (2015) popularized the method by using a temperature-based softmax. Since then, KD has blossomed into an active area of research.

Logit-Based KD algorithms (Hinton et al., 2015; Zhao et al., 2022; Yang et al., 2021; Chen et al., 2022) look at the logits, the input to the classification softmax. Feature-based algorithms (Romero et al., 2014; Zagoruyko & Komodakis, 2016; Ahn et al., 2019; Passalis & Tefas, 2018; Heo et al., 2019b; Kim et al., 2018; Huang & Wang, 2017; Heo et al., 2019a; Chen et al., 2021a;b; Miles et al., 2021; Srinivas & Fleuret, 2018; Tian et al., 2020) probe the middle layers of both networks and take different perspectives on how intermediate knowledge should be quantified and transferred. This leads to variable performance due to the engineering tricks in choosing and connecting layers, especially when there is a significant difference between the teacher and student architectures. To circumvent these problems, relation-based methods (Park et al., 2019; Liu et al., 2021; Yim et al., 2017; Tung & Mori, 2019; Peng et al., 2019) look at the inter-example relationships between outputs of the teacher and student networks; for example, we can ensure that the pairwise distances between all features output by the student are equal to those of the teacher. These have been shown to be highly effective, though at a higher computational cost since, for a batch of $n$ inputs, $2 * \binom{n}{2}$ distances must be computed for the outputs of both networks. Li et al. (2022) proposed an Expectation-Maximization (EM) approach to dynamically focus on the best samples to transfer knowledge. woo Kwak et al. (2022) showed that KD could be useful in the active learning regime as well.

The target performance we aim to beat, a pre-trained student, has rarely been addressed. Most prior work looked at distilling all of ImageNet (Deng et al., 2009; Russakovsky et al., 2015) and finetuning the student after to test task performance. In the NLP literature, this 'foundation transfer' is extensively used to shrink the size of large language models like BERT (Sanh et al., 2019), thereby creating a smaller foundation model. KDEP (He et al., 2022a) used KD for pre-training and only used a random $10\%$ of ImageNet. Fundamentally, these approaches are just stronger forms of pre-training the student model. In addition, cross-architecture teacher-student pairings, e.g. transformer-to-convolutional, have seldom been tried due to the inflexibility of certain KD losses to differing implicit biases induced by model architectures.

The closest prior work with regards to the use of a pre-trained teacher can be found in the works of Srinivas & Fleuret (2018) and Ahn et al. (2019). They both considered the case of using a pre-trained teacher, with no classification head, distilling its knowledge of an unseen task to a student. However, these works viewed the situation as a form of *transfer learning*, where the teacher never touches the downstream task. We believe neither work focused enough on just how significant the performance gains from a pre-trained teacher could be.

### 2.2 CONTRASTIVE LEARNING

Contrastive Learning comes from the idea of Noise Contrastive Estimation (NCE) (Gutmann & Hyvärinen, 2010). Broadly speaking, the main idea is to pull "positive pairs" together, and push "negative pairs" far apart. Early iterations of this idea (Chopra et al., 2005; Schroff et al., 2015; Sohn, 2016; Salakhutdinov & Hinton, 2007; Frosst et al., 2019; Oord et al., 2018) dealt with different sources and quantities for positive and negative pairs.

Most modern advancements in contrastive learning come from the realm of self-supervised learning. The positive pair is often 2 augmented versions of the same sample, while the negatives come from other samples. SimCLR (Chen et al., 2020a) took the rest of the mini-batch as negatives but consequently required large batch sizes to work effectively. MoCo (He et al., 2020; Chen et al., 2020b) used a memory queue to hold previous samples that get passed into a momentum updated copy of the model. Other approaches such as BYOL (Grill et al., 2020), SimSiam (Chen & He, 2021), and Barlow Twins (Zbontar et al., 2021) completely sidestep the use of negative samples, though it has been shown that some of these implicitly use them by design (Fetterman & Albrecht, 2020).

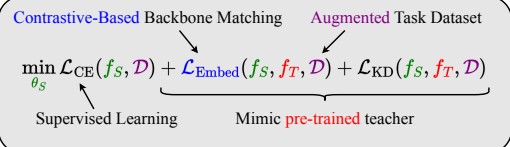

| Method | High Accuracy | Low Cost |
|---|---|---|
| Supervised | ✗ | ✓ |
| Pre-training | ✓ | ✗ |
| DPT (Ours) | ✓ | ✓ |

Figure 2: (Left) Our proposed alternative to pre-training and finetuning. (Right) The benefits of our method compared to standard approaches.

## 2.3 Generative Models as Data

Due to the increased realism of synthetic images, recent work has examined if they are good enough to use as a source of data since we can *infinitely* sample from generative models. Ideally, if a generative model perfectly models the data distribution, then we can gather more samples than our dataset provides. In the past, GANs (Goodfellow et al., 2014) were the de-facto generative model. Nowadays, diffusion models (Sohl-Dickstein et al., 2015; Ho et al., 2020), equivalent to score-based methods (Hyvärinen, 2005; Vincent, 2011; Song & Ermon, 2019; Song et al., 2020) offer better generative guarantees in terms of sampling diversity and likelihood maximization. The only downside to diffusion models, as of writing, is their slow sampling speed. While GANs take one neural network evaluation to generate a batch of samples, diffusion models take $T$ (sometimes thousands of) evaluations to denoise. Improving sampling speeds with minimal effect on sample quality is an active area of research (Song et al., 2023; Salimans & Ho, 2022).

The works of He et al. (2022b); Zhou et al. (2023); Bansal & Grover (2023); Sariyildiz et al. (2022); Azizi et al. (2023) used diffusion to augment the training dataset for supervised training with positive results, but did not consider the KD setting. Liu et al. (2018) applied the idea of generated samples to KD; however, they trained a GAN to model the task distribution, whereas we use a pre-trained foundation model that never touches our task distribution.

## 3 Method

Our main idea is to regularize a small model's supervised learning process for task $\mathcal{T}$ by having it match the behavior of a "teacher" model $f_T$, which has been pre-trained on a large dataset and finetuned for $\mathcal{T}$. In doing so, the target small "student" model $f_S$ behaves as if it too was pre-trained and finetuned on $\mathcal{T}$, when in reality, it never touches the pre-training dataset. This paradigm involves three design choices: (1) the choice of $f_T$, (2) the knowledge transfer loss function $\mathcal{L}$, and (3) the choice of the transfer dataset $\mathcal{D}$. $f_S$ then learns its parameters $\theta_S$ in this setup (Figure 2). Our focus on the benefits achieved via optimizing all three of these aspects sets us apart from prior work.

### 3.1 A Teacher $f_T$ With the Desired Backbone

In order for $f_S$ to behave like it has a pre-trained backbone, it should learn from a model that has one. Thus, we propose a regularizing teacher $f_T$, which can simply be an off-the-shelf, pre-trained model that is finetuned for the desired downstream task we wish $f_S$ to learn. To reflect modern practices, when we refer to finetuning, we imply the method of *linear probing* (LP) which freezes the backbone weights and only updates the task-specific head. Aside from the computational benefits, this also avoids distorting the backbone (Kumar et al., 2022).

It is possible for our teacher to be worse than one that was trained end-to-end from scratch. In particular, linearly probing can lead to *inferior* performance compared to vanilla training due to distribution shifts between the pre-training and downstream datasets. We find that *this is not a problem*. Surprisingly, even when a teacher's classification head becomes suboptimal due to the shift, $f_S$ still achieves strong performance (Section 4.2). In other words, $f_S$ can be assisted by a teacher that is *cheaper* to train (compared to training end-to-end from scratch) yet ends up performing *better*. We attribute this to the design of our loss function $\mathcal{L}$, which provides teacher supervision at both the logit and feature level.

```
# x: batch of samples
# y: batch of other positive half

import torch
def align_loss(x, y, alpha=2):
    dists = (x - y).norm(p=2, dim=1)
    return dists.pow(alpha).mean()

def unif_loss(x, t=2):
    pair = -t*torch.pdist(x, p=2)**2
    return pair.exp().mean().log()
```

**Require:** $f_S, f_T, g_S, g_T$: models and projectors
**Require:** $w_a, w_u$: loss weights
1: Sample batch $x$
2: $z_S \leftarrow g_S \circ f_S(x), z_T \leftarrow g_T \circ f_T(x)$
3: $l_a = \texttt{align\_loss}(z_S, z_T)$
4: $l_u = \frac{1}{2}(\texttt{unif\_loss}(z_S) + \texttt{unif\_loss}(z_T))$
5: $\mathcal{L}_{\text{Embed}} = w_a l_a + w_u l_u$

(a) Pseudocode

(b) Flow chart

Figure 3: Our Alignment/Uniformity (A/U) based contrastive loss.

## 3.2 THE LOSS FUNCTION $\mathcal{L}$

Since we linearly probe the teacher, its backbone contains the untouched knowledge of the foundation dataset. Thus, we want to ensure the student's backbone mimics a pre-trained one on the desired task. To avoid the complexities induced by the implicit biases of model architectures and encourage generalizability, we look to contrastive learning theory, which only utilizes the final embeddings $h$.

### 3.2.1 THE CONTRASTIVE-BASED EMBEDDING LOSS

**Contrastive-Based Distillation** For the sake of simplicity, in this section we will ignore the classification heads and view the student $f_S$ and teacher $f_T$ as models that generate high-dimensional representations $h_S \in \mathbb{R}^S, h_T \in \mathbb{R}^T, S \neq T$ after their final backbone layer (Figure 3b). We believe a robust measure of distance can be effective given its recent success in the self-supervised domain: contrastive learning. Following the ideas from their literature, we append "projection" modules $g_S : \mathbb{R}^S \to \mathbb{R}^d, g_T : \mathbb{R}^T \to \mathbb{R}^d$, where $d \ll S, T$. Thus, given a data distribution $x \sim p_{\text{data}}$, our models induce distributions $p_S(\cdot), p_T(\cdot)$, such that $g_S \circ f_S(x) =: z_S \sim p_S, g_T \circ f_T(x) =: z_T \sim p_T$. Note that both $g(\cdot)$'s include a final normalization step that ensures $||z||_2 = 1$. We also define the distribution of "positive" pairs $p_{\text{pos}}(\cdot, \cdot)$, where the marginals should match: $\forall z_S, \int p_{\text{pos}}(z_S, z_T)dz_T = p_S(z_S)$ and $\forall z_T, \int p_{\text{pos}}(z_S, z_T)dz_S = p_T(z_T)$. We approximate sampling from $p_{\text{pos}}(\cdot, \cdot)$ by passing the same sample $x_i \sim p_{\text{data}}$ through both networks and projectors to generate the positive pair $(z_{S,i}, z_{T,i})$. *Any* sample $x_j, j \neq i$ that creates a $z_{\cdot,j}$ induces negative pairs $(z_{S,i}, z_{T,j}), (z_{S,j}, z_{T,i})$.

At its core, contrastive learning pulls positive pairs together and pushes negative pairs apart. Our proposed formulation that achieves this, inspired by the InfoNCE loss of Oord et al. (2018), is to minimize the following with respect to the parameters of the student $f_S$ and both projectors $g_S, g_T$:

$$\mathbb{E}_{(z_{S,i}, z_{T,i}) \sim p_{\text{pos}}, \{z_{S,j}\}_{j=1}^M \sim p_S, \{z_{T,j}\}_{j=1}^M \sim p_T} \left[ -\log \frac{e^{z_{S,i}^\top z_{T,i}/\tau}}{\sum_j e^{z_{S,i}^\top z_{T,j}/\tau} + \sum_j e^{z_{S,j}^\top z_{T,i}/\tau}} \right] \quad (1)$$

Our modernized variant allows us to bridge knowledge distillation and any contrastive-based learning algorithms. We choose one that is inexpensive and interpretable as an illustration, leaving other contrastive-distillation collaborations to future work.

**Optimizing Alignment and Uniformity**  Wang & Isola (2020) showed that as the number of negative samples $M \to \infty$, then 2 simpler quantities can be optimized instead: the *alignment* (cosine similarity) of the positive pairs, and the *uniformity* of all (normalized) samples on the hypershpere in $\mathbb{R}^d$. With no negative sample bank, momentum model, or large batch size, this method is the most lightweight contrastive algorithm. Thus, we chose to this Alignment/Uniformity ((**A/U**)) as the core of our contrastive-distillation algorithm.

We re-express their alignment metric to reflect our distillation flavor of generating positive pairs:

$$\mathcal{L}_{\text{align}} := \mathbb{E}_{(z_{S,i}, z_{T,i}) \sim p_{\text{pos}}}[||z_{S,i} - z_{T,i}||_2^\alpha], \alpha > 0 \tag{2}$$

The uniformity metric is expressed as follows, where $Z \in \{S, T\}$:

$$\mathcal{L}_{\text{uniform}} := E_{(z_{Z,i}, z_{Z,j}) \sim p_Z}[e^{-t||z_{Z,i} - z_{Z,j}||_2^2}], t > 0 \tag{3}$$

Putting these together, we obtain a loss term that is a weighted combination of Equations 2 and 3: $\mathcal{L}_{\text{Embed}} := w_{\text{align}} \cdot \mathcal{L}_{\text{align}} + w_{\text{uniform}} \cdot \mathcal{L}_{\text{uniform}}$. We use the default parameters suggested in Wang & Isola (2020): $w_{\text{align}} = 1, w_{\text{uniform}} = 1, \alpha = 2, t = 2$. Pseudocode can be found in Figure 3a.

**A Contrastive Formulation with a Memory Bank**  Tian et al. (2020) were the first to apply the idea of contrastive learning to KD via CRD, a contrastive loss inspired by Wu et al. (2018), though their work pre-dated most modern advancements. CRD fits into Equation 1 by using an artificially larger $M$ supplemented by extra negative samples $z_{S_j}, z_{T_j}$ kept in a momentum-updated memory bank. To show the flexibility of our formulation, we additionally try a modernized version of their method, which we call **CRD++**.

### 3.2.2 THE LOGIT LOSS

While not necessary, we can additionally employ a logit-based loss given its high effectiveness and low cost for knowledge transfer. The original method of Hinton et al. (2015) compares the student and teacher logits (pre-softmax vectors) via the cross-entropy:

$$\mathcal{L}_{KD}(z_s, z_t) = \tau^2 \mathcal{H}(\sigma(z_t/\tau), \sigma(z_s/\tau)) \tag{4}$$

where $\sigma$ is the softmax function and $\tau$ is a temperature hyperparameter that controls the smoothness of the softmax; a higher $\tau$ leads to a more uniform distribution.

### 3.2.3 FINAL LOSS FUNCTION

Our final loss function combines the above loss functions with the standard cross-entropy loss $\mathcal{L}_{\text{CE}}$ on the true one-hot labels: $\mathcal{L} := \lambda_1 \mathcal{L}_{\text{CE}} + \lambda_2 \mathcal{L}_{\text{Embed}} + \lambda_3 \mathcal{L}_{\text{KD}}$. We used $\lambda_1 = \lambda_2 = \lambda_3 = 1$. Ablations for the overall A/U design can be found in Appendix A.3.

### 3.3 AUGMENTING THE TRANSFER DATASET

Lastly, we can choose the transfer dataset $\mathcal{D}$. We avoid touching the pre-training dataset due to the high costs, but *only* using the task dataset is suboptimal. Fortunately, we can leverage recent results in generative modelling to solve this dilemma.

### 3.3.1 THE DATA GAP

First, let's take a step back and recall what makes pre-training so effective. The theory of transfer learning rests on the belief that by seeing enough diverse data, an encoder $f$ becomes an excellent feature extractor that can generalize to unseen datasets. Observing this phenomenon from the perspective of the sample space, we can attribute the success to the possibility that in high enough dimensions, the domains of many downstream tasks span subspaces that can be "handled" by the learned manifold. In other words, learning enough of this generic manifold leads to successful interpolation to explicitly unexplored subsets of the domain.

By forgoing the pre-training dataset, the target model misses out on this benefit. Its learned manifold can only leverage information learned from the task; unseen parts of the domain, e.g. the test dataset, are only handled well if there's enough overlap with the training task. Thus, we propose the existence of a *data gap*, which describes the difference between the task dataset and the parts of the pre-training dataset which help the model generalize at test-time. If we can fill in this gap, performance should increase. Assuming this perspective, we propose to augment the transfer dataset $\mathcal{D}$ with synthetic samples for the desired task (see Figure 1).

### 3.3.2 SOURCE OF SYNTHETIC SAMPLES

Given the recent successes of diffusion models in generative modelling, we choose to use a pre-trained text-to-image model, Stable Diffusion (Rombach et al., 2022), as our source of extra samples. Diffusion models (Sohl-Dickstein et al., 2015; Ho et al., 2020; Song et al., 2020), destroy an input distribution into an isotropic Gaussian and learn the reverse denoising process to generate samples. When combined with an autoencoder $\mathcal{E}$ that projects samples to a manageable space, a network $\epsilon_\theta(z_t, t)$ is trained to predict the noise to remove at step $t$: $\mathbb{E}_{\mathcal{E}(x), \epsilon \sim \mathcal{N}(0,1), t} \left[ ||\epsilon - \epsilon_\theta(z_t, t)||_2^2 \right]$.

We leverage the publicly available stable-diffusion-v1-4[1], which was pre-trained on LAION-2B (Schuhmann et al., 2022), and guide its generation with the appropriate language prompt (Table 1). We found these hand-chosen prompts to work well, and leave more advanced strategies to future work. Note the cost of this approach: we avoid training the generative model, but generation via the current state of diffusion models takes hundreds of sequential model evaluations.

## 4 EXPERIMENTS

### 4.1 SETUP

In our results, we use several abbreviations for brevity. **FR** stands for a model that was initialized randomly and trained end-to-end on the task. **LP** refers to a model that was pre-trained on ImageNet (Deng et al., 2009; Russakovsky et al., 2015) and linearly probed for the task.

Table 1: Prompts fed into Stable Diffusion to generate images. {class} corresponds to a class name in the dataset.

| Dataset | Prompt |
|---|---|
| MIT-67 | "the inside of a {class}" |
| DTD | "{class} texture" |
| Caltech-101 | "a picture of a {class}" |

**T** and **S** refer to the teacher and student models, respectively. Lastly, **A/U-[FR/LP]** refers to our knowledge transfer algorithm that uses a FR or LP teacher, respectively.

Details about optimizers, schedulers, and input augmentations can be found in Appendix A.5.

**Datasets** We choose data-limited datasets since they are most helped by pre-training: MIT-67 (Quattoni & Torralba, 2009), CUB-2011 (Wah et al., 2011), DTD (Cimpoi et al., 2014), and Caltech-101 (Li et al., 2003). In addition, we evaluate on standard vision benchmarks: CIFAR-10 and CIFAR-100 Krizhevsky et al. (2009). The details of these can be found in Appendix A.6.

**Models** We test our method with 2 teachers: a ResNet50 (He et al., 2016) and a base Vision Transformer (Dosovitskiy et al., 2020), ViT-B-16. For the target small models, we choose a ResNet18 and a MobileNetV2 (Sandler et al., 2018).

### 4.2 FRESH VS. PRE-TRAINED TEACHER

To test for the effect of the teacher choice, we fixed $\mathcal{L}_{\text{Embed}}$ to be our A/U loss and kept $\mathcal{D}$ as the task dataset. The teacher was a ResNet50 and the student was a MobileNetV2.

The results can be found in Figure 4. Interestingly, even when linear probing the teacher leads to *worse* downstream performance (see the results for CIFAR-10 and CIFAR-100), distilling from T-LP is still a good idea as it's less costly to train on the teacher's side and leads to competitive, and often better, performance on the student side. To test whether a weaker classification head can hurt the student's learning process, we also tested omitting the logit-based loss on CIFAR-10 and CIFAR-100 in Appendix A.4.

---

[1] https://huggingface.co/CompVis/stable-diffusion-v1-4

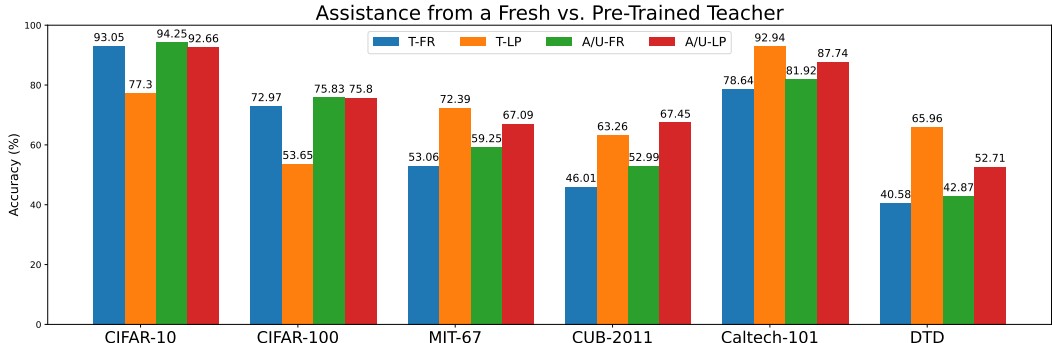

Figure 4: Regardless of whether a teacher trained from scratch (T-FR) or pretrained-then-finetuned (T-LP) performs better on the downstream task, assistance from a finetuned teacher (A/U-LP) is competitive with, and often outperforms, assistance from a fresh teacher (A/U-FR).

## 4.3 TESTING THE LOSS FUNCTION

Given the superior performance of pre-trained teachers, we test our loss function using them. We keep $\mathcal{D}$ as the task dataset for now.

**Comparison to Other Methods** For a fair comparison, we also test using a pre-trained teacher with prior state-of-the-art KD algorithms that did not consider our setting. Notably, we choose ones that have demonstrated high efficacy as well as being architecture-agnostic; they do not utilize any intermediate feature information. The three works we compare to are KD (Hinton et al., 2015), an updated version of CRD (Tian et al., 2020) which we denote CRD++, and SRRL (Yang et al., 2021). Details regarding these can be found in Appendix A.7.

**Results** The results can be found in Table 2 and in Appendix A.2 for other architecture pairings. In all setups, our algorithm (A/U) leads to high performance gains that are competitive with, and sometimes surpass, prior work. On one hand, these results demonstrate the superiority of our A/U loss in certain setups; on the other hand, many existing KD algorithms can be substituted in $\mathcal{L}$ for our overall paradigm and still offer competitive boosts to small model performance. We also emphasize that, while CRD++ is also a contrastive method, our A/U method is computationally cheaper due to the lack of a negative sample bank, while often outperforming it. Lastly, we additionally compare our method to self-supervised approaches in Appendix A.1; we find that both A/U and CRD++ vastly outperform prior work, and ours again benefits from using less resources.

Table 2: Accuracies (%) of MobileNetV2 when assisted by a pre-trained ResNet50.

| Method | CIFAR-10 | CIFAR-100 | MIT-67 | CUB-2011 | Caltech-101 | DTD |
|---|---|---|---|---|---|---|
| T-LP | 77.3 | 53.65 | 72.39 | 63.26 | 92.94 | 65.96 |
| S-FR | 92.83 | 72.39 | 57.84 | 58.23 | 81.13 | 44.47 |
| KD | 92.92 | 75.97 | 66.04 | **67.48** | 86.61 | 52.02 |
| SRRL | 92.85 | 73.24 | 65.67 | 62.81 | 85.31 | 49.79 |
| CRD++ | **93.45** | **76.37** | 65.75 | 67.21 | 86.89 | 52.45 |
| **A/U (Ours)** | 92.66 | 75.8 | **67.09** | 67.45 | **87.74** | **52.71** |

## 4.4 BENEFITS OF AN AUGMENTED TRANSFER DATASET

Finally, we combine all our proposed improvements by adding the synthetic samples, dubbing the entire method **DPT**. To test the overall efficacy, we compare to a pre-trained *student* that is linearly probed on the task dataset. We denote augmenting the transfer dataset with synthetic samples as "DPT ($n\times$)", where $n\times$ indicates the number of extra samples in terms of the size of the respective

Table 3: Our method (DPT) can achieve performance close to, and often surpass, pre-training.

| Method | CIFAR-10 | CIFAR-100 | MIT-67 | CUB-2011 | Caltech-101 | DTD |
|---|---|---|---|---|---|---|
| DPT (0×) | **92.66** | **75.8** | 67.09 | **67.45** | 87.74 | 52.71 |
| DPT (1×) | - | - | 71.57 | - | 89.94 | 59.68 |
| DPT (2×) | - | - | **72.54** | - | 89.83 | 60.21 |
| S-LP | 74.84 | 52.08 | 68.13 | 65.14 | **91.02** | **62.29** |

train set (see Appendix A.6). In addition, we apply standard image augmentations, e.g. manipulating color properties, cropping, and flipping. The results on the ResNet50-MobileNetV2 pair can be found in Table 3.

On half of the datasets, we surpass a pre-trained student without needing to augment $\mathcal{D}$. For the remaining three, one succeeded when adding either (1×) or (2×) synthetic samples. For the remaining two, they get close: only lagging behind by around 1-2%.

## 4.5 Cost Analysis

The main practical benefit of our method is the time saved by avoiding pre-training. We showcase our method in Figure 5. The cost of DPT ($n\times$) includes the times to finetune the teacher and generate the images. Our method can cut training time by up to 94%, though gets slower if more samples need to be generated. We do not include the time to pre-train the teacher or generative models since those are assumed to be obtained already trained off-the-shelf.

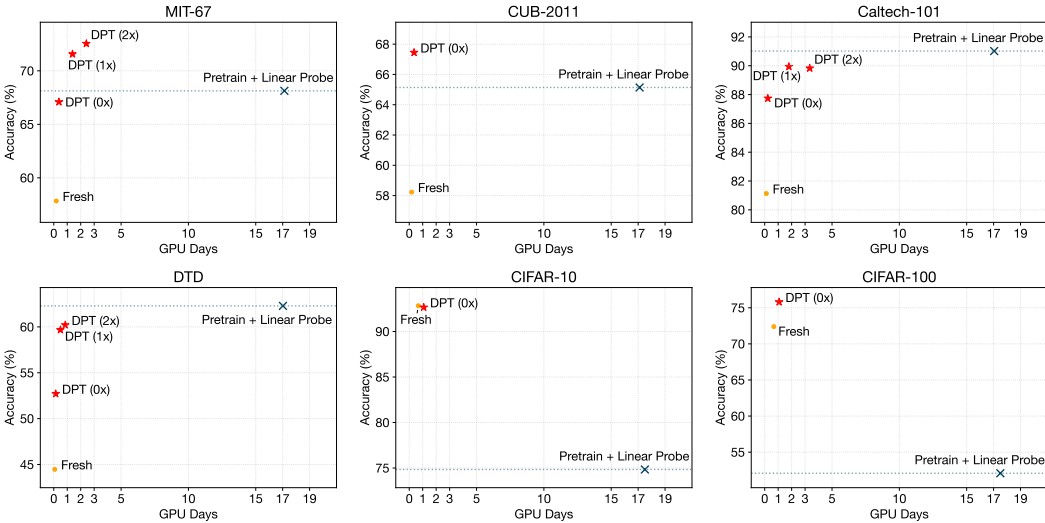

Figure 5: Cost/accuracy comparison of our method to supervised training (Fresh) and pre-training then linear probing. The teacher was a ResNet50, the student was a MobileNetV2, and we used our A/U loss for regularization. All timing experiments were done on one NVIDIA P100 GPU.

## 5 Conclusion

Mainstream attention continues to focus on large models. Fortunately, we show that small models can leverage these advancements for their own benefit. By combining a pre-trained teacher, a novel knowledge transfer algorithm, and augmenting the "questions" the student asks the teacher, *any* small model can significantly improve its performance without weeks of pre-training; all that is required is *any* publicly available, pre-trained teacher, contrastive learning algorithm, and generative model. When combined with methods like NAS and pruning, small models have a bright future.

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

# A    APPENDIX

## A.1    SELF-SUPERVISED TRANSFER LEARNING

In the self-supervised learning literature, pre-training is significantly less effective as model size decreases. SEED (Fang et al., 2021), DisCo (Gao et al., 2021), and BINGO (Xu et al., 2021) attempt to mitigate this by augmenting training with distillation from a pre-trained teacher backbone. They test performance by training/distilling on ImageNet first, then adapting to tasks. We compare our strategy to theirs in Table 4. We use the same teacher model, a pre-trained MoCoV2 ResNet50, and student, a ResNet18. The baseline is a ResNet18 pre-trained on ImageNet via MoCoV2 and linearly probed for a task. We omit the teacher's head to match their teacher-student setup ($\lambda_3 = 0$), so the knowledge transfer is purely feature-based. Our method surpasses prior work by a large margin, lending support to the idea that small models may not need their own strong backbone, especially in the self-supervised domain where model size matters more.

Table 4: Comparing to self-supervised distillation.

| Method | CIFAR-10 | CIFAR-100 |
|---|---|---|
| Baseline | 77.9 | 48.1 |
| SEED (Fang et al., 2021) | 82.3 | 56.8 |
| DisCo (Gao et al., 2021) | 85.3 | 63.3 |
| BINGO (Xu et al., 2021) | 86.8 | 66.5 |
| CRD++ | **94.44** | **75.46** |
| A/U (Ours) | 94.33 | 73.83 |

## A.2    ADDITIONAL MODEL PAIRINGS

In the ViT-MobileNetV2 (Table 5), ResNet50-ResNet18 (Table 6), and ViT-MobeilNetV2 (Table 7) setups, our A/U loss is often the best of second-best method.

Table 5: Accuracies (%) of MobileNetV2 when assisted by a pre-trained ViT-B-16.

| Method | CIFAR-10 | CIFAR-100 | MIT-67 | CUB-2011 | Caltech-101 | DTD |
|---|---|---|---|---|---|---|
| T-LP | 95.14 | 80.56 | 81.34 | 79.06 | 94.80 | 70.21 |
| S-FR | 92.83 | 72.39 | 57.84 | 58.23 | 81.13 | 44.47 |
| KD | 94.33 | 78.9 | **66.34** | **71.16** | 84.92 | **49.79** |
| SRRL | 93.99 | 78.66 | 63.51 | 67.66 | **85.31** | 49.73 |
| CRD++ | 93.93 | 78.57 | 65.37 | 70.45 | 82.49 | 48.4 |
| **A/U (Ours)** | **94.55** | **79.71** | 65.97 | 70.87 | 84.8 | 49.47 |

## A.3    ABLATIONS ON CONTRASTIVE LOSS DESIGN

Our ablation experiments were done in the process of designing the A/U loss formulation. We tested whether the original KD loss (Hinton et al., 2015) or the newer SRRL loss (Yang et al., 2021) should

Table 6: Accuracies (%) of ResNet18 when assisted by a pre-trained ResNet50.

| Method | CIFAR-10 | CIFAR-100 | MIT-67 | CUB-2011 | Caltech-101 | DTD |
|---|---|---|---|---|---|---|
| T-LP | 77.3 | 53.65 | 72.39 | 63.26 | 92.94 | 65.96 |
| S-FR | 93.22 | 71.44 | 49.85 | 49.17 | 77.40 | 34.47 |
| KD | 92.48 | **75.94** | 65.22 | **64.77** | 85.14 | 46.44 |
| SRRL | **92.93** | 72.06 | 64.63 | 62.74 | **86.27** | 47.82 |
| CRD++ | 92.87 | 75.72 | 65.75 | 64.19 | 85.25 | **48.78** |
| **A/U (Ours)** | 92.79 | 75.11 | **66.72** | 63.88 | 85.54 | 46.01 |
| S-LP | 78.38 | 55.44 | 66.27 | 62.75 | 90.23 | 62.07 |

Table 7: Accuracies (%) of ResNet18 when assisted by a pre-trained ViT-B-16.

| Method | CIFAR-10 | CIFAR-100 | MIT-67 | CUB-2011 | Caltech-101 | DTD |
|---|---|---|---|---|---|---|
| T-LP | 95.14 | 80.56 | 81.34 | 79.06 | 94.80 | 70.21 |
| S-FR | 93.22 | 71.44 | 49.85 | 49.17 | 77.40 | 34.47 |
| KD | 94.54 | 78.95 | **65.82** | **68.81** | 83.28 | **44.15** |
| SRRL | 94.03 | 77.46 | 61.87 | 62.94 | 81.58 | 37.66 |
| CRD++ | 93.94 | **79.06** | 63.28 | 67.35 | 79.49 | 42.61 |
| **A/U (Ours)** | **94.98** | 78.89 | 65.75 | 68.48 | **83.39** | 40.69 |
| S-LP | 78.38 | 55.44 | 66.27 | 62.75 | 90.23 | 62.07 |

be used for the logit-based loss. In addition, we experimented with the design of the projection module $g(\cdot)$. The teacher-student pairing was ResNet50-MobileNetV2. The results can be found in Table 8.

**$\mathcal{L}_{\text{KD}}$ vs. SRRL Loss**   In SRRL (Yang et al., 2021), instead of comparing the student and teacher logits, the student features are first passed through a connector module (Figure 8) that lifts them to the teacher's feature space then passed through the teacher's classifier. Then, these logits are compared. When comparing using this or $\mathcal{L}_{\text{KD}}$ for the logit-based loss, we found that $\mathcal{L}_{\text{KD}}$ worked better.

**Projector Design**   In CRD (Tian et al., 2020), the projection module to $\mathbb{R}^d$ is a linear matrix multiplication. Since its publication, improvements on the contrastive learning regime have been proposed, one of which is the use of a *deeper* projector (Chen et al., 2020b;a). Thus, we also compared using the 2-layer MLP architecture found in MoCoV2 (Chen et al., 2020b) vs. a linear one. The exact design can be found in Figure 6. For our A/U based loss, we found that both the deeper and linear one achieved good performance but chose the deeper one to show the flexibility of our design. Our source code provides different options for the choice of the projector. Substituting the deeper projector in CRD exactly describes CRD++.

Table 8: Ablation on different projector architectures and logit-based losses for our A/U loss.

| Projector | Logit Loss | CIFAR-10 | CIFAR-100 | MIT-67 | CUB-2011 | Caltech-101 | DTD |
|---|---|---|---|---|---|---|---|
| MoCoV2 | SRRL | **93.25** | 73.29 | 65.82 | 63.03 | 87.23 | 49.63 |
| MoCoV2 | KD | 92.66 | **75.8** | 67.09 | 67.45 | **87.74** | **52.71** |
| Linear | SRRL | 93.18 | 73.16 | 66.19 | 62.08 | 86.95 | 50.74 |
| Linear | KD | 92.81 | **75.8** | **67.54** | **68.16** | 87.51 | 52.07 |

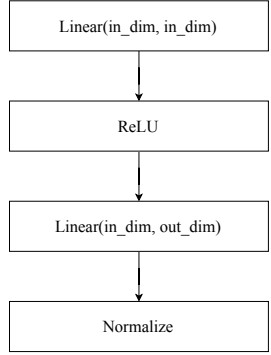

Figure 6: The design of the projector $g(\cdot)$ in A/U and CRD++.

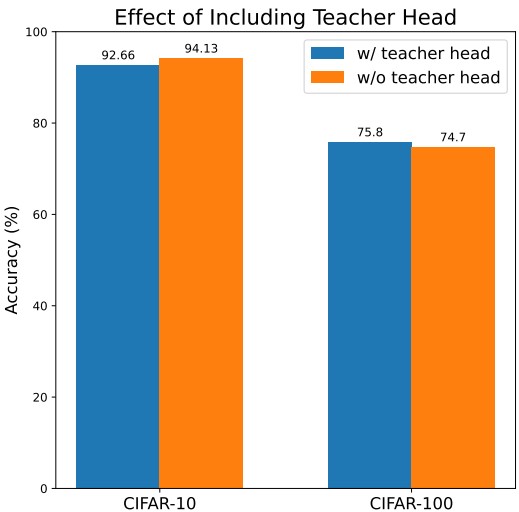

Figure 7: When using a finetuned teacher, the classification head may be weaker than one that was trained end-to-end due to distribution shift. We find that, in these cases, omitting this head during knowledge transfer has minimal impact on the student's final accuracy.

## A.4  OMITTING A BAD CLASSIFICATION HEAD

In certain cases, linear probing leads to inferior performance compared to training end-to-end from scratch. We observed this phenomenon when training a ResNet50 on CIFAR-10 and CIFAR-100, so we tested whether keeping this head could hurt the student's learning process. The results can be found in Figure 7, which show that in these cases, keeping the head does not sacrifice performance.

## A.5  EXPERIMENT DETAILS

For training, linear probing, and distilling, we used the AdamW optimizer (Loshchilov & Hutter, 2019) with the default parameters: 1e-3 initial learning rate, beta = (0.9, 0.999), epsilon=1e-8 and a weight decay of 0.01. No additional learning rate schedulers were used. All experiments were done for 250 epochs with a random seed of 9. Inputs were centered, resized to 224x224, and randomly flipped horizontally. Pre-trained models were taken from PyTorch's model hub [2].

---

[2]https://pytorch.org/vision/stable/models.html

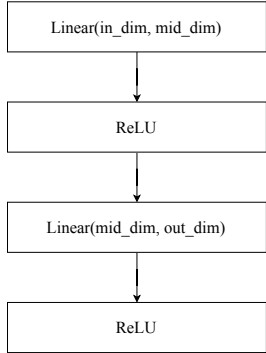

Figure 8: Our modified design of the module that projects the student features to the teacher dimensions in SRRL. mid_dim is the average of in_dim and out_dim.

## A.6 DATASET DETAILS

We used the Indoor Scene Recognition dataset (MIT-67) Quattoni & Torralba (2009), the Caltech-UCSD Birds-200-2011 dataset (CUB-2011) Wah et al. (2011), the Describable Textures Dataset (DTD) Cimpoi et al. (2014), and CIFAR-10/100 Krizhevsky et al. (2009). The first 4 are considered data-limited. MIT-67 has 67 classes with 5360 images in the train set. CUB-2011 has 200 classes with 5994 images in its train set. Caltech-101 has 6907 train images across 101 classes, though some classes have many more (hundreds) of images while others have as low as 30. DTD has a train set of size 1880 for 47 classes. Lastly, CIFAR-10 and CIFAR-100 are standard vision datasets with 50,000 training images across 10 and 100 classes, respectively.

## A.7 DETAILS OF OTHER KD METHODS

1. KD (Hinton et al., 2015): The original KD loss that compares logits.
2. CRD++ (Tian et al., 2020): A contrastive-based distillation method. We also include $\mathcal{L}_{KD}$. As their work pre-dated many modern advancements in contrastive learning, we compare our method to a modernized version which substitutes the linear projection $g(\cdot)$ with the same 2-layer MLP we use in our A/U loss.
3. SRRL (Yang et al., 2021): A method that passes the student features through the teacher classifier. We do not use $\mathcal{L}_{KD}$ here as the original authors did not. However, they only considered convolutional networks and utilized a small convnet to project the student features to the teacher's dimension. We substitute this with an MLP to make it general to the underlying model architecture (Figure 8).

The implementations of these methods were taken from their open-source repositories, which we thank the authors for making available, and adapted appropriately.

