# OpenReview forum: "Don't Pre-train, Teach Your Small Model"
_ICLR.cc/2024/Conference — Submitted to ICLR 2024_

### Official Review · Reviewer_r4D4 · 2023-10-26

**Soundness:** 3 good
**Presentation:** 3 good
**Contribution:** 2 fair
**Rating:** 5
**Confidence:** 3

**Summary:**

The paper proposes a way to effectively train smaller models when pretrained model for particular architecture is not available off-the-shelf.  There are three main components to their approach (1) fine-tune teacher model on desired task using linear probing (2) use contrastive learning to distill specific knowledge from teacher to student model (3) use artificially generated samples using generative models to get performance boost. They demonstrate that the above approach is beats student model pretrained and linearly probed both in terms of accuracy and GPU days.

**Strengths:**

- Combines different approaches in literature to create a recipe for training smaller models in the era of large models.
- Proposes a contrastive learning based training technique which might be helpful when we don't have pretrained models available for certain specific architectures.
- The paper is well written and easy to understand.

**Weaknesses:**

- Baselines considered for experiments seems weaker. I would have liked to see the comparison between student model which is pretrained and fully fine tuned vs DPT (2x).
- Datasets considered in the paper are such that we can easily use generative model to create more data. The approach might not give good performance in cases where we don't get good images using generative models.

**Questions:**

- From a practical standpoint where do you think this method might be helpful? Because of the models considered in the paper have pre-trained counterparts available off-the-shelf i.e cost of pretraining is already paid and you only need to fine tune these models to downstream tasks.
- In this case synthetic data generation is a key step in boosting performance. In some datasets generative models might not be able to do good job in adding more synthetic data. Would the approach usefulness fail here?

---

> ### Author Response · Authors · 2023-11-15
>
> 1. A fully finetuned student would be an interesting data point. Intuitively, we believe that it would achieve better performance, but would be the most expensive method on our cost plots. In addition, this would really limit the student to being good on only one task due to the backbone distortion mentioned in section 3.1. However, if you believe that this data would significantly strengthen the main intention of our paper, we can provide it.
> 2. From a practical standpoint, we believe it’s useful for those who want to use a small model that *isn’t* available in any of these hubs. Granted, we used small models that were already trained on ImageNet in our illustrative examples, but that was mainly for our convenience and research efficiency. However, the future of model hubs and licenses is unknown, but our method enables practitioners to lower their reliance on hubs for all of their model needs.
> 3. *Sometimes* data generation is a key step. Importantly, we find that the synthetic data component is the most significant in the data scarce scenarios. As we can see (Table 3), the datasets with plenty of samples (CIFAR-10, CIFAR-100, CUB-2011) are able to surpass their pre-trained counterparts without adding any synthetic data (DPT (0x)). Overall, synthetic data just boosts performance at test time, which is an observation known to the community for a while now. If it is the case that the synthetic data is not as easily generated, then our method would take a small hit, but by no means would it be failing as a result. Echoing what we said in the Novelty section above, the synthetic data should be viewed as a free benefit of work in pre-trained deep generative models. If it is there and works; great. If not, one still has the teacher and distillation algorithm that can give significant gains regardless.

---

### Official Review · Reviewer_YmUb · 2023-10-30

**Soundness:** 3 good
**Presentation:** 3 good
**Contribution:** 1 poor
**Rating:** 1
**Confidence:** 5

**Summary:**

This paper proposes distillation from a pretrained model finetuned on a target task at hand alongside training on additional synthetic data from a large generative model in order to train a smaller, equi- or even higher-performant model. They also propose a variation of existing contrastive distillation motivated by the uniformity & alignment objective proposed in Wang et al. 2020.
Leveraging large-scale pretrained models both as teacher and for data generation, the authors show strong performance gains for smaller base models (MobileNetV2 & ResNet18).

**Strengths:**

The paper is overall well structured and presented, being consequently easy to parse and understand.

**Weaknesses:**

I have several large issues with this work, which primarily stem from the lack of novelty.

* In particular, the proposed scenario, in which one distills from a teacher model which was pretrained ahead of time before being adapted to the target task at hand, is essentially just the standard distillation setting with the ONLY difference being that the teacher starts from pre-training.

* Are there any significant differences in the insights gained or the difficulties surrounding the distillation process that require such a separate treatment (beyond the fact that distillation from teachers that were pretrained ahead of time is in itself not novel)?

* Furthermore, the proposed distillation objective appears very derivative of contrastive distillatio proposed in Tian et al. 2020. It would be important if the authors provided a stronger differentation here.

* Similarly, the use of synthetic data to help train a model is orthogonal and not novel, and has been studied to significant extend, particularly with recent improvements in large-scale generative modeling as the authors also note in their related works. However, just deploying this to the task of Knowledge Distillation is an insufficient contribution.

* Beyond the lack of novelty, the experiments are also unfortunately quite lackluster - for a claim that involves applicability to "any pretrained teacher model", only testing on two teachers does not provide sufficient breadth. In addition, I'm not sure if the provided results are comparable - is synthetic data also used for the base finetuning of the teacher models, which are then claimed to be outperformed by the student using large-scale synthetic data?

* Adding to that, just training a linear probe on top of a teacher is insufficient exploration into teacher finetuning to claim sufficient adaptation to a target task at hand, particular when distribution shifts are larger.

* Finally, I have some issues regarding formulations used in this paper, starting at the title iself, which is contradictory to what is actually proposed, as the introduced setup does leverage pretraining TO teach and for synthetic data generation. Beyond that, the authors sell knowledge distillation as the primary means for suitable model deployment on edge devices to leverage smaller architectures, disregarding research into model quantization, pruning, etc. It would be great if this could be contextualized better.

**Questions:**

In order for me to raise my score, the issues with respect to the lack of novelty as listed above have to be addressed, alongside a discussion about the significance and relevance of the experimental results.

---

> ### Author Response · Authors · 2023-11-15
>
> 4. (1, 2, 4) addressed in Novelty above
> 5. (3) With regards to the distillation algorithm itself, as we mentioned in the paper, our choice of the alignment/uniformity metric was purely illustrative. We appreciate the work of CRD in bridging contrastive learning with KD, but their work was done pre- the modern boom in contrastive learning brought on by SimCLR, MoCo, CLIP, etc. Therefore, we formulate KD as a contrastive objective with the modern notation and considerations, massaging it into an InfoNCE-type form that SimCLR and MoCo build off of. From here, any recent development (most of which took place after the publication of CRD), could be used, and which we assume would also be successful. The A/U metric was chosen since one of the main themes of the paper was efficiency; if we’re going to minimize the training time for the small model, let’s try a method that doesn’t include all the bells and whistles such as large batch sizes or momentum-updated models.
> 6. [5] We welcome any suggestions for other models you believe would make our case stronger. We believed that choosing a popular convo-based one and transformer-based one was sufficient to demonstrate that our theory was justified in practice. Most other vision models are just derivatives of these 2 model philosophies. In addition, the synthetic data is never seen by the teacher during training (thought this is a good idea!), and we added clarifying language for this. We do not believe this is cause for concern, because for 3 datasets (CIFAR-10, CIFAR-100, and CUB-2011), we are able to demonstrate that without any synthetic samples our paradigm works to beat the baseline.
> 7. (6) Our reasons for linear probing as opposed to full finetuning come from theoretical and practical considerations. While full finetuning is certainly more robust to distribution shifts, it has been found to lead to a distortion of the backbone which will probably harm future finetuning to other tasks. We wish to leave our teacher “unharmed” in the case where we wish to use the foundation for multiple student tasks. From a practical perspective, most foundations that are finetuned for a task in the modern era are linearly probed for this exact reason. In addition, full finetuning incurs a *much* higher cost on the practitioners end as they now have to train the teacher end-to-end on each task. With the increasing size of foundations, this would be a significant cost for the intended users of our method. This is fundamentally even closer to the “vanilla KD” regime that you bring up in your first point; one minor improvement we have is that by taking an off-the-shelf pre-trained one, we only need to train one linear layer and our teacher is ready to provide big improvements.
> 8. (7) This sounds like a grammatical issue, which we don’t agree is contradictory to the message of our paper. We believe it is reasonable for readers to infer that the “Pre-train” in the title refers to the object of the phrase, “small model”, as opposed to being a declaration of not pre-training anything ever. This should also be clear after reading the abstract and/or introduction as well. For how it fits in the broader context of other methods, see above “Comparison to NAS, Pruning, Quantization”.
> 9. For the datapoints requested: (1) the situation where the teacher is exposed to the synthetic data and (2) where it is fully finetuned, we believe that these would just lead to better performance altogether, but would not significantly alter our conclusions. However, if you disagree, we can attempt to provide them but may not be able to gather all the data by the deadline.

---

> > ### Comment · Reviewer_YmUb · 2023-11-16
> >
> > I thank the authors for the replies. After reading their response, I still have several
> > issues with this work. In particular:
> >
> > ---
> >
> > __Point [4.]__
> > I do not believe that the shared response answers this convincingly. Is it not true that the primary proposal in this paper is to leverage distillation from a teacher that wasn't trained from scratch to some distillation datasets, but rather pretrained? If so, I do not see any novelty in the conclusions that are drawn here - it is still distillation from a stronger teacher, with the minor difference that it was pretrained. It seems that the authors agree on this aspect as per the shared response. But given that, it remains unclear to me what the actual contribution in this paper is. Because it is not the insight that you can distill from a stronger teacher, and it can't be the fact that training on synthetic data benefits model performance or allows useful representation to be learned (see e.g. [1,2,3,4]), particularly when leveraging generative models that were trained on much more expansive datasets. And using both together in my eyes is not sufficient contribution.
> >
> > [1] Beery at al. 2019, "Synthetic Examples Improve Generalization for Rare Classes"
> > [2] Lehner et al. 2023, "3D Adversarial Augmentation for Robust Out-of-Domain Predictions"
> > [3] Tian et al. 2023, "StableRep: Synthetic Images from Text-to-Image Models Make Strong Visual Representation Learners"
> > [4] Azizi et al. 2023, "Synthetic Data from Diffusion Models Improves ImageNet Classification"
> >
> > ---
> >
> > __Point [6.]__
> > > We welcome any suggestions for other models (...). Most other vision models are just derivatives of these 2 model philosophies.
> >
> > I still maintain that for such a general statement, investigating only two particular architectures is insufficient. There are many other popular architectures, and the simple use of convolution or attention does not equate all architectures. Different structures elicit different optimization behavior, and changes in size incur significant changes in generalization.
> >
> > ---
> >
> > __Point [7.]__
> > > From a practical perspective, most foundations that are finetuned for a task in the modern era are linearly probed (...).
> >
> > It would be great if the authors provided clear references for this statement, as especially recently, low-ranking adaptation has grown significantly as means to emulate full finetuning in a much more cost-efficient regime. Linear probing only allows for very limited adaptation, in which one already assumes forms of linear separability of downstream samples by the pretrained model. This does not hold for general train-test distribution shifts. Generally, as per my understanding, linear probing has seen primary usage in the domain of self-supervised learning to evaluate the applicability of learned representations on the same pretraining domain. But this does not mean that linear probing is used or recommended as a general tool for any form of downstream adaptation.
> > As such, I would definitely appreciate further references and details here, particularly for the mentioned theoretical considerations.
> >
> > ---
> >
> > __Point [8.]__
> > I was not referring to any grammatical issues, but rather to the issue of implicit pretraining by simply having the pretraining stage incorporated into the use of large-scale pretrained teacher models.

---

### Official Review · Reviewer_vMzu · 2023-11-01

**Soundness:** 3 good
**Presentation:** 3 good
**Contribution:** 2 fair
**Rating:** 3
**Confidence:** 4

**Summary:**

This paper proposes a new method on training small models without pre-training them on large datasets. The authors first use a pre-trained teacher model that is fine-tuned for the desired task and regularise the student model to match its behaviour via a contrastive-based knowledge distillation loss. Showing that the proposed method, called Don’t Pre-train, Teach (DPT), the authors claim that it can achieve similar or better performance than pre-training and finetuning while saving significant resources and time.

**Strengths:**

1. The paper is well-written and easy to understand.

2. The method proposed in this paper is technically sound to me.

**Weaknesses:**

**1. Absence of crucial references and an insufficient comparison with state-of-the-art methods in knowledge distillation.**

While the author has touched upon various studies in the domain of knowledge distillation, the discussion appears to be incomplete. For example, works such as [1, 2] that delve into efficient (pre-)training via knowledge distillation techniques, have already revealed that knowledge distillation can be used as an effective technique to boost the training efficiency for tasks including image classification, object detection and semantic segmentation.

In a closely related domain, active knowledge distillation can, to a degree, achieve the objectives set by the authors, making it imperative to discuss and compare with studies like [3].

**2. The observation offered by the authors in this paper is not novel to me.**

Considering these pertinent studies listed above, the main contribution of this paper, *i.e.* employing distillation for expediting the training process, doesn't strike me as particularly novel. One new aspect of this paper is the authors' use of solely synthetic data for distillation. However, this shift in data source doesn't seem substantial enough to stand as the core contribution for a paper published at ICLR.

**3. The experimental results presented in this paper are somewhat underwhelming to me.**

In light of the benchmarks set by leading methods in the realm of knowledge distillation, the performance depicted in this paper feels somewhat middling. For instance, when employing ResNet-18 as the student and ResNet-50 as the teacher, [1] manages to attain 82.22% in a mere 16 GPU hours. In contrast, this paper's proposed technique takes over 24 GPU hours to reach a modest 75.8%.

While I concede that there might be nuances that make a direct comparison slightly skewed, it remains necessary for the authors to validate their approach in comparable scenarios and surpass at least one of the aforementioned studies. One experiment that might be needed is to merge the real and synthetic data and then apply the proposed method of this paper to see whether the proposed method can surpass the previous works in knowledge distillation.


### Reference

> [1] He R, Sun S, Yang J, et al. Knowledge distillation as efficient pre-training: Faster convergence, higher data-efficiency, and better transferability, CVPR 2022.

> [2] Li C, Lin M, Ding Z, et al. Knowledge condensation distillation. ECCV 2022.

> [3] Kwak B, Kim Y, Kim Y J, et al. Trustal: Trustworthy active learning using knowledge distillation. AAAI 2022.

**Questions:**

See weaknesses.

---

> ### Author Response · Authors · 2023-11-15
>
> 1. We added references to these works, thank you for pointing them out. However, we do not believe a comparison to them is appropriate. KDEP is fundamentally a pre-training optimization, whereas we want to completely avoid pre-training of any kind. KCD takes an E-M approach to the choice of the transfer dataset each iteration, whereas we propose to augment it at the beginning. Both of these are great ideas, and there is no reason why one cannot apply them to our formulation as well (i.e. running KCD on a synthetically-augmented transfer set, or doing KDEP to warm up the student, although this loses the spirit of our approach). We believe both these approaches tackle different aspects of the training procedure that do not intersect with our work. TrustAL is an interesting approach, though their work concerns the language domain. More importantly, we do not see how the problem that they tackle, example forgetting, is related to our work that justifies a comparison. Sure, they use a teacher model to help minimize forgotten knowledge, but that is fundamentally a different use of a teacher model compared to ours. Again, there is an opportunity here to utilize their method in tandem with ours in an active learning setup.
> 2. Addressed above in Novelty section
> 3. KDEP is a great work that proposes a well-justified and effective optimization to pre-training. We particularly enjoyed their observation that entire foundation datasets may not even be necessary to pre-train! However, we maintain the position that a direct comparison to their work is unnecessary. In addition, the “16 hours” claimed is not a comparable data point without accounting for differences in hardware used. Their paper states that their timing numbers were done on 4 V100 GPU’s, whereas we used 1 P100, an older and slower architecture. Even if we ignore the supremacy of the Volta (V) architecture over Pascal (P), this translates to KDEP taking around 64 hours to achieve that 82%, which is 3x longer than our method; however, they do achieve a higher accuracy at this high cost. We believe that the desired “comparable scenario” is addressed in Section 4.3, where we fix the same teacher-student, use the same transfer dataset, but compare to popular and effective distillation algorithms: vanilla KD, CRD, and SRRL. There, we see that our method is competitive, but no method comes out to be the outstanding winner. Since our main contribution is the overall recipe (see Novelty response above), we are not looking to show purely the supremacy of our contrastive-based distillation algorithm, but that we can formulate KD as a contrastive objective, and any algorithm (like the Align/Uniform we choose) can be competitive to SoTA. Also, our experiments do indeed combine the real and synthetic datasets. DPT (1x) combines the real dataset with an equally sized synthetic one, while DPT (2x) combines the real with a synthetic one twice the size. If there is anywhere in the paper that we can clarify this better, please let us know.

---

### Official Review · Reviewer_FsBG · 2023-11-01

**Soundness:** 2 fair
**Presentation:** 4 excellent
**Contribution:** 2 fair
**Rating:** 3
**Confidence:** 3

**Summary:**

This paper revisits the optimal approach to training small models and proposes a middle ground between traditional supervised training from scratch and the resource-intensive pre-training followed by finetuning. By regularizing the feature backbone of a model being trained from scratch to align with an existing pre-trained model, and using a knowledge distillation loss rooted in the theory of contrastive learning, the authors demonstrate good performance for small models across six computer vision datasets.

**Strengths:**

* The question of effectively training small models is crucial in the drive for resource-efficient methodologies. This paper's approach is commendable for addressing this challenge.
* The proposed training method is based on the solid theory and is well embedded into related work. Comparison to recent SoTA is highly appreciated.

**Weaknesses:**

* While the technical progress in efficient model training is appreciated, the delta to existing works is rather small, leading to limited novelty.
* The claim of general applicability might benefit from a more diverse testing. Relying on only 6 computer vision datasets and a handful of teacher-student architectures narrows the perspective.
* The variability in the results, as seen in Fig. 4 and Tab. 2, 3, 5, and 6, makes it challenging to draw clear conclusions. The results provided in the appendix (Tab. 5 and 6) show better performance of CRD++. S-LP is not defined in the text.
* It would be helpful to have clarity on the number of experiments behind each reported measurement and the inclusion of standard deviations.
* "small models" and cost considerations need to be clearly defined. The paper seems not to account for certain factors that contribute to the total cost, such as training the existing pre-trained models.
* How does the method compare to NAS or pruning (given limited space or a constraint on the inference time)? The reviewer believes the problem of training a "small model" for a specific domain needs a more holistic approach. Considering both learning methods and architectural choices, might offer a more comprehensive solution for edge and mobile devices. The proposed method looks at the problem in a limited way by considering training / data aspects only.
* While synthetic samples do enhance performance, it remains unclear if baselines are also exposed to them during training, raising questions about fairness in comparisons.

**Questions:**

See weaknesses.

---

> ### Author Response · Authors · 2023-11-15
>
> 1. Addressed in Novelty section above
> 2. We welcome suggestions as to other datasets or pairings you believe would strengthen our case. Granted, we chose 1 modality, but in this modality we attempted to use datasets that reflect different realities (data is abundant vs data is scarce). For the models, we carefully chose pairings to exemplify how flexible our method is. One conv-based and one transformer-based teacher have no problems working with small models that have different design approaches.
>  3. The conclusion to be taken from the tables is that the overall flow is highly effective and multiple distillation methods can be used to get around similar levels of performance. Vanilla KD can’t always be used (e.g. self-supervised regime, Table 4 in Appendix A.1), and many new KD algorithms tend to over-engineer, in our opinion, the knowledge transfer. Instead, our generic contrastive formulation is straightforward, makes no assumptions about the underlying model architectures, and performs competitively to previous SoTA works. We find that most modern works in KD that are accepted to similar venues propose some new heuristic distillation algorithm and test it on certain model pairings that show their superiority. However, when subjecting just 2 of these, CRD and SRRL, to reproduction in our paper, we find that while they perform well, none are overall the “best”. We wish for attention to focus on the bridge between contrastive learning and distillation; practitioners can then decide which contrastive algorithm they wish to try and empirically find to work best. In addition, S-LP is defined in section 4.1, where we claim that “S” refers to the student, and “LP” refers to pre-trained on ImageNet and linearly probed for the task.
> 4. We only did 1 experiment for each. More experiments are always better, but given our time and resources, we do not believe we can run every experiment multiple times by the deadline. However, a future version of this paper would definitely have them.
> 5. If by “pre-training cost”, you are referring to the cost of pre-training the teacher and generative models, we chose to omit those since those costs are not absorbed by the intended practitioners of this method. They can assume that these models already exist and are available for the using immediately. We added a clarification about this in our updated version.
> 6. Addressed in Comparison section above
> 7. The baselines were not exposed to the synthetic samples. The only scenarios that were were DPT (1x) and DPT (2x). We updated the wording in our results section to clarify that only the transfer dataset is augmented. We can certainly provide those if desired, but does the reviewer believe that these would lead to a different or stronger conclusion than is currently provided?

---

### Author Response · Authors · 2023-11-15

Hello. We thank all reviewers for their insight and appreciate the time and care that went into the responses and suggestions to improve our work. Since there was overlap between reviews, we will respond to the larger themes here, and leave more individual responses in the respective threads.

**Novelty**

Re: concerns about a lack of novelty. If viewing our work from the same perspective as these reviews, we, in fact, agree. However, the novelty we wish for readers to walk away with is not the “ingredients”, but the “recipe”; we believe that, as concurrent advancements take place in 3 different lines of work: large pre-trained foundations, contrastive learning, and generative models, the small-model regime significantly benefits from these *for free*. One just needs to combine them in a particular flow to achieve better results at a much lower cost than the standard pre-training flow. In fact, we believe our work sheds light on the reality that the efforts in large model improvements can directly improve small models at the same time. We believe these observations have flown under the radar and can be appreciated by the broader research and practice community.

Upon re-reading our submission, we realized that our presentation did not reflect this desired takeaway as effectively as it could have. We have provided an updated version with wording changes that gets this idea across better. Most changes occurred in the Abstract and Introduction, though minor changes can be found in the Method, Experiments, and Conclusion.

**Comparison to NAS, Pruning, and Quantization**

Small model optimization is not only limited to the realm of knowledge distillation. We are cognizant of the impressive advancements in other approaches such as NAS, pruning, and quantization. However, we believe a comparison to these methods isn’t entirely appropriate. The primary reason is because we see these not as points of comparison, but as future opportunities for practitioners to utilize these methods in tandem with ours. NAS is fundamentally a paradigm that takes place *before* training, pruning and quantization take place *after* training, and our method fits right in the middle *during* training. One reviewer mentioned KDEP, which is, at its core, a pre-training optimization. These approaches are not mutually exclusive to ours but can be combined in a straightforward manner. For example, one can begin by choosing an architecture developed by NAS, run our paradigm, then prune and/or quantize it. We believe a comparison between these methods would promote the wrong idea that one has a choice *between* these methods, as opposed to each being an independently useful tool at different points in the development process.

On another note, all of these methods incur tremendous additional costs. One of the largest benefits of our method is how lightweight it is. The most expensive additional operation is generating synthetic samples, and the research community is heavily focused on improving this aspect as of writing.

---

> ### Comment · Reviewer_YmUb · 2023-11-16
>
> I thank the authors for their reply, and will respond to the more general statements here first.
>
> ---
>
> __On the novelty__
> _(This is a copy of my individual reply)_.
> I do not believe that the shared response answers this convincingly. Is it not true that the primary proposal in this paper is to leverage distillation from a teacher that wasn't trained from scratch to some distillation datasets, but rather pretrained? If so, I do not see any novelty in the conclusions that are drawn here - it is still distillation from a stronger teacher, with the minor difference that it was pretrained. It seems that the authors agree on this aspect as per the shared response. But given that, it remains unclear to me what the actual contribution in this paper is. Because it is not the insight that you can distill from a stronger teacher, and it can't be the fact that training on synthetic data benefits model performance or allows useful representation to be learned (see e.g. [1,2,3,4]), particularly when leveraging generative models that were trained on much more expansive datasets. And using both together in my eyes is not sufficient contribution.
>
> [1] Beery at al. 2019, "Synthetic Examples Improve Generalization for Rare Classes"
> [2] Lehner et al. 2023, "3D Adversarial Augmentation for Robust Out-of-Domain Predictions"
> [3] Tian et al. 2023, "StableRep: Synthetic Images from Text-to-Image Models Make Strong Visual Representation Learners"
> [4] Azizi et al. 2023, "Synthetic Data from Diffusion Models Improves ImageNet Classification"
>
> ---
>
> __On the comparison to NAS / Pruning / Quantization__
> > ... all these methods incur tremendous additional costs. One of the largest benefits of our method is how lightweight it is.
>
> I don't agree with this statement - generating hundreds of thousands synthetic examples, with comparable number of forward passes through the foundation model to distill from is anything but lightweight. Indeed, from a cost perspective, I'd wager pruning to be notably cheaper - though I'm not 100% sure, and this is where the issue of lacking comparisons to these existing approaches comes into play.
> A direct comparison would give the reader much more intuition into *when* and *why* the proposed approach should be applied. In my eyes, it isn't even a performance competition, different approaches may operate orthogonally. But without any comparisons, no clear statements can be made in that regard.

---

### Meta-Review · Area_Chair_2mQz · 2023-12-08

**Metareview:**

This paper proposes a strategy to train a small model, that involves knowledge distillation of a pretrained teacher model with a novel theoretically-grounded contrastive loss and synthetic data. The experimental results show that the proposed scheme is effective over training the model from scratch, or the popular pretraining then finetuning strategy.

The paper received unanimously negative reviews, due to the lack of novelty in the strategy and the components involved. Basically, the strategy of distilling the pretrained model into smaller models is a common practice, and what are different in this work are the use of synthetic data and the new distillation loss. However, even the latter two components lacks novelty as there exist prior works on data-free distillation and contrastive feature distillation. Although the authors argued the novelty in the way they combined the existing techniques to make their framework work, the reviewers remained unconvinced about the novelty.

Another important issue brought up by the reviewers is the lack of experimental comparison against pruning or NAS methods, which renders the effectiveness and the practical impact of the proposed work somewhat inconclusive.

Therefore the paper does not seem ready for publication in its current state.

**Justification For Why Not Higher Score:**

The proposed framework and methodological components lacks novelty, and also there is missing comparison against relevant works from pruning or NAS.

**Justification For Why Not Lower Score:**

N/A

---

### Decision · Program_Chairs · 2024-01-16

Reject